# Entropy of Quantum Measurements

**DOI:** 10.3390/e24111686

**Published:** 2022-11-18

**Authors:** Stanley Gudder

**Affiliations:** Department of Mathematics, University of Denver, Denver, CO 80208, USA; sgudder@du.edu

**Keywords:** entropy, quantum measurements, effects, observables

## Abstract

If *a* is a quantum effect and ρ is a state, we define the ρ-entropy $S_{a}\left(\rho \right)$ which gives the amount of uncertainty that a measurement of *a* provides about ρ. The smaller $S_{a}\left(\rho \right)$ is, the more information a measurement of *a* gives about ρ. In Entropy for Effects, we provide bounds on $S_{a}\left(\rho \right)$ and show that if a+b is an effect, then $S_{a+b}\left(\rho \right)\geq S_{a}\left(\rho \right)+S_{b}\left(\rho \right)$. We then prove a result concerning convex mixtures of effects. We also consider sequential products of effects and their ρ-entropies. In Entropy of Observables and Instruments, we employ $S_{a}\left(\rho \right)$ to define the ρ-entropy $S_{A}\left(\rho \right)$ for an observable *A*. We show that $S_{A}\left(\rho \right)$ directly provides the ρ-entropy $S_{I}\left(\rho \right)$ for an instrument I. We establish bounds for $S_{A}\left(\rho \right)$ and prove characterizations for when these bounds are obtained. These give simplified proofs of results given in the literature. We also consider ρ-entropies for measurement models, sequential products of observables and coarse-graining of observables. Various examples that illustrate the theory are provided.

## 1. Introduction

In an interesting article, D. Šafránek and J. Thingna introduce the concept of entropy for quantum instruments [1]. Various important theorems are proved and applications are given. In quantum computation and information theory one of the most important problems is to determine an unknown state by applying measurements on the system [2,3,4,5]. Entropy provides a quantification for the amount of information given to solve this so-called state discrimination problem [6,7,8]. In this article, we first define the entropy for the most basic measurement, namely a quantum effect *a* [2,3,9,10]. If ρ is a state, we define the ρ-entropy $S_{a}\left(\rho \right)$ which gives the amount of uncertainty (or randomness) that a measurement of *a* provides about ρ. The smaller $S_{a}\left(\rho \right)$ is, the more information a measurement of *a* provides about ρ. In Section 2, we give bounds on $S_{a}\left(\rho \right)$ and show that if a+b is an effect then $S_{a+b}\left(\rho \right)\leq S_{a}\left(\rho \right)+S_{b}\left(\rho \right)$. We then prove a result concerning convex mixtures of effects. We also consider sequential products of effects and their ρ-entropies.

In Section 3, we employ $S_{a}\left(\rho \right)$ to define the entropy $S_{A}\left(\rho \right)$ for an observable *A*. Then $S_{A}\left(\rho \right)$ gives the uncertainty that a measurement of *A* provides about ρ. We show that $S_{A}\left(\rho \right)$ directly gives the ρ-entropy $S_{I}\left(\rho \right)$ for an instrument I. We establish bounds for $S_{A}\left(\rho \right)$ and characterize when these bounds are obtained. These give simplified proofs of results given in [1,5,11]. We also consider ρ-entropies for measurement models, sequential products of observables and coarse-graining of observables. Various examples that illustrate the theory are provided. In this work, all Hilbert spaces are assumed to be finite dimensional. Although this is a restriction, the work applies for quantum computation and information theory [2,3,9,10].

## 2. Entropy for Effects

Let *H* be a finite dimensional complex Hilbert space with dimension *n*. We denote the set of linear operators on *H* by L(H) and the set of states on *H* by S(H). If ρ∈S(H) with nonzero eigenvalues $\lambda_{1},\lambda_{2},\ldots ,\lambda_{m}$ including multiplicities, the *von Neumann entropy* of ρ is [4,6,7,8].
$$S\left(\rho \right)=-\sum_{i=1}^{m}\lambda_{i}\ln\left(\lambda_{i}\right)=-\mathrm{tr}\;\left[\rho \ln(\rho )\right]$$
We consider S(ρ) as a measure of the randomness or uncertainty of ρ and smaller values of S(ρ) indicate more information content. For example, ρ is the completely random state I/n, where *I* is the identity operator, if and only if S(ρ)=ln(n) and ρ is a pure state if and only if S(ρ)=0. Moreover, it is well-known that 0≤S(ρ)≤ln(n) for all ρ∈S(H). The following properties of *S* are well-known [4,6,8]:$$\begin{array}{cc} S(U\rho U^{*}) & =S(\rho )\;\mathrm{when}\;U\;\mathrm{is}\;\mathrm{unitary}\\ S(\rho_{1}⊗\rho_{2}) & =S\left(\rho_{1}\right)+S\left(\rho_{2}\right)\\ \sum \mu_{i}S\left(\rho_{i}\right) & \leq S\left(\sum \mu_{i}\rho_{i}\right)\leq \sum \mu_{i}S\left(\rho_{i}\right)-\sum \mu_{i}\ln\left(\mu_{i}\right) \end{array}$$
where $0\leq \mu_{i}=1$ with $\sum \mu_{i}=1$.

An operator a∈L(H) that satisfies 0≤a≤I is called an *effect* [2,3,9,10]. We think of an effect *a* as a two-outcome yes-no measurement. If a measurement of *a* results in outcome yes we say that *a occurs* and if it results in outcome no then *a does not occur*. The effect $a^{\prime }=I-a$ is the *complement* of *a* and $a^{\prime }$ occurs if and only if *a* does not occur. We denote the set of effects by E(H). If a∈E(H) and ρ∈S(H) then 0≤tr(ρa)≤1 and we interpret tr(ρa) as the probability that *a* occurs when the system is in state ρ. If a≠0 we define the ρ-*entropy* of *a* to be
(1)$$S_{a}\left(\rho \right)=-\mathrm{tr}\;\left(\rho a\right)\ln\left[\frac{\mathrm{tr}\;(\rho a)}{\mathrm{tr}\;(a)}\right]$$
We interpret $S_{a}\left(\rho \right)$ as the amount of uncertainty that the system is in state ρ resulting from a measurement of *a*. The smaller $S_{a}\left(\rho \right)$ is, the more information a measurement of *a* gives about ρ. Such information is useful for state discrimination problems [2,3,4,5].

If ρ is the completely random state I/n then (Equation 1) becomes
$$S_{a}\left(I/n\right)=-\mathrm{tr}\;\left(Ia/n\right)\ln\left[\frac{\mathrm{tr}\;(Ia/n)}{\mathrm{tr}\;(a)}\right]=-\frac{1}{n}\;\mathrm{tr}\;\left(a\right)\ln\left(\frac{1}{n}\right)=\frac{\mathrm{tr}\;(a)}{n}\;\ln\left(n\right)$$
Since tr(a)≤n we conclude that $S_{a}\left(I/n\right)\leq S\left(I/n\right)$ for all a∈E(H). Another extreme case is when a=λI for 0<λ≤1. We then have for any ρ∈S(H) that
$$S_{\lambda I}\left(\rho \right)=-\mathrm{tr}\;\left(\rho \lambda I\right)\ln\left[\frac{\mathrm{tr}\;(\rho \lambda I)}{\mathrm{tr}\;(\lambda I)}\right]=-\lambda \ln\left[\frac{\lambda }{\lambda \mathrm{tr}\;(I)}\right]=\lambda \ln\left(n\right)$$
Thus, as λ gets smaller, the more information we gain.

A real-valued function with domain D(f), an interval in R, is *strictly convex* if for any $x_{1},x_{2}\in D\left(f\right)$ with $x_{1}\neq x_{2}$ and 0<λ<1 we have
$$f\left[\lambda x_{1}+\left(1-\lambda \right)x_{2}\right]<\lambda f\left(x_{1}\right)+\left(1-\lambda \right)f\left(x_{2}\right)$$
If the opposite inequality holds, then *f* is *strictly concave*. It is clear that *f* is strictly convex if and only if −f is strictly concave. Of special importance in this work are the strictly convex functions −lnx and xlnx. We shall frequently employ Jensen’s theorem which says: if *f* is strictly convex and $0\leq \mu_{i}\leq 1$ with $\sum_{i=1}^{m}\mu_{i}=1$, then
$$f\left(\sum_{i=1}^{m}\mu_{i}x_{i}\right)\leq \sum_{i=1}^{m}\mu_{i}f\left(x_{i}\right)$$
Moreover, we have equality if and only if $x_{i}=x_{j}$ for all i,j=1,2,…,m [1].

**Theorem 1.** 
*If ρ∈S(H) with nonzero eigenvalues $\lambda_{i}$, i=1,2,…,m, and a∈E(H) with tr(ρa)≠0, then*

$$-\sum_{i}\mathrm{tr}\;\left(P_{i}a\right)\lambda_{i}\ln\left(\lambda_{i}\right)\leq S_{a}\left(\rho \right)\leq \ln\left[\frac{\mathrm{tr}\;(a)}{\mathrm{tr}\;(\rho a)}\right]$$

*where $\rho =\sum_{i}\lambda_{i}P_{i}$ is the spectral decomposition of ρ. Moreover, $S_{a}\left(\rho \right)=\ln\left[\mathrm{tr}\;(a)/\mathrm{tr}\;(\rho a)\right]$ if and only if tr(ρa)=1 in which case $S_{a}\left(\rho \right)=ln\left[\mathrm{tr}\;(a)\right]$ and if*

(2)
$$S_{a}\left(\rho \right)=-\sum_{i}\mathrm{tr}\;\left(P_{i}a\right)\lambda_{i}\ln\left(\lambda_{i}\right)$$

*then $\mathrm{tr}\;\left(P_{i}a\right)=\mathrm{tr}\;\left(P_{j}a\right)$ for all i,j=1,2,…,m and $S_{a}\left(\rho \right)=\left(\mathrm{tr}\;\left(a\right)/m\right)S\left(\rho \right)$ while if $\mathrm{tr}\;\left(P_{i}a\right)=\mathrm{tr}\;\left(P_{j}a\right)$ for all i,j=1,2,…m then $S_{a}\left(\rho \right)=\left(\mathrm{tr}\;\left(a\right)/m\right)\ln\left(m\right)$.*


**Proof.** Letting $\mu_{j}=\mathrm{tr}\;\left(P_{j}a\right)/\mathrm{tr}\;\left(a\right)$, j=1,2,…,m, we have that $0\leq \mu_{j}\leq 1$ and $\sum_{j}\mu_{j}=1$. Since −xln(x) is strictly concave we obtain
$$\begin{array}{cc} S_{a}\left(\rho \right) & =-\mathrm{tr}\;\left(\rho a\right)\ln\left[\frac{\mathrm{tr}\;(\rho a)}{\mathrm{tr}\;(a)}\right]=-\mathrm{tr}\;\left(\sum_{i}\lambda_{i}P_{i}a\right)\ln\left[\frac{\mathrm{tr}\;\left(\sum_{j}\lambda_{j}P_{j}a\right)}{\mathrm{tr}\;(a)}\right]\\ \; & =-\sum \lambda_{i}\mathrm{tr}\;\left(P_{i}a\right)\ln\left(\sum_{j}\lambda_{j}\mu_{j}\right)=\mathrm{tr}\;\left(a\right)\left[-\sum_{i}\lambda_{i}\mu_{i}\left(\sum_{j}\lambda_{j}\mu_{j}\right)\right]\\ \; & \geq -\mathrm{tr}\;\left(a\right)\sum_{i}\mu_{i}\lambda_{i}\ln\left(\lambda_{i}\right)=-\mathrm{tr}\;\left(a\right)\sum_{i}\frac{\mathrm{tr}\;(P_{i}a)}{\mathrm{tr}\;(a)}\;\lambda_{i}\ln\left(\lambda_{i}\right)\\ \; & =-\sum_{i}\mathrm{tr}\;\left(P_{i}a\right)\lambda_{i}\ln\left(\lambda_{i}\right) \end{array}$$
Since
$$\mathrm{tr}\;\left(\rho a\right)=\mathrm{tr}\;\left(a^{1/2}\rho a^{1/2}\right)\leq \mathrm{tr}\;\left(\rho \right)=1$$
we have that
$$S_{a}\left(\rho \right)=\mathrm{tr}\;\left(\rho a\right)\ln\left[\frac{\mathrm{tr}\;(a)}{\mathrm{tr}\;(\rho a)}\right]\leq \ln\left[\frac{\mathrm{tr}\;(a)}{\mathrm{tr}\;(\rho a)}\right]$$
If tr(ρa)=1, then
$$S_{a}\left(\rho \right)=-\mathrm{tr}\;\left(\rho a\right)\ln\left[\frac{\mathrm{tr}\;(\rho a)}{\mathrm{tr}\;(\rho a)}\right]=-\ln\left[\frac{1}{\mathrm{tr}\;(a)}\right]=\ln\left[\mathrm{tr}\;(a)\right]$$
Conversely, if $S_{a}\left(\rho \right)=\ln\left[\mathrm{tr}\;(a)/\mathrm{tr}\;(\rho a)\right]$, then clearly tr(ρa)=1. If (Equation 2) holds, then we have equality for Jensen’s inequality. Hence, $\mathrm{tr}\;\left(P_{i}a\right)=\mathrm{tr}\;\left(P_{j}a\right)$ for all i,j=1,2,…,m. Since
$$\mathrm{tr}\;\left(a\right)=\sum_{i}\mathrm{tr}\;\left(P_{i}a\right)=m\mathrm{tr}\;\left(P_{i}a\right)$$
we conclude that
$$S_{a}\left(\rho \right)=-\mathrm{tr}\;\left(P_{1}a\right)\sum_{i}\lambda_{i}\ln\left(\lambda_{i}\right)=\frac{\mathrm{tr}\;(a)}{m}\;S\left(\rho \right)$$
Finally, suppose $\mathrm{tr}\;\left(P_{i}a\right)=\mathrm{tr}\;\left(P_{j}a\right)$ for all i,j=1,2,…,m. Then
$$\mathrm{tr}\;\left(a\right)=\sum_{i}\mathrm{tr}\;\left(P_{i}a\right)=m\mathrm{tr}\;\left(P_{1}a\right)$$
We conclude that
$$\begin{array}{cc} S_{a}\left(\rho \right) & =-\mathrm{tr}\;\left(P_{1}a\right)\sum_{i}\lambda_{i}\ln\left[\sum_{j}\lambda_{j}\frac{\mathrm{tr}\;(P_{1}a)}{\mathrm{tr}\;(a)}\right]=-\mathrm{tr}\;\left(P_{1}a\right)\sum_{i}\lambda_{i}\ln\left(\sum_{j}\lambda_{j}\frac{1}{m}\right)\\ \; & =-\mathrm{tr}\;\left(P_{1}a\right)\sum_{i}\lambda_{i}\ln\left(\frac{1}{m}\right)=\frac{\mathrm{tr}\;(a)}{m}\;\ln\left(m\right)e \end{array}$$ □

For a,b∈E(H) we write a⊥b if a+b∈E(H).

**Theorem 2.** 
*If a⊥b, then $S_{a+b}\left(\rho \right)\geq S_{a}\left(\rho \right)+S_{b}\left(\rho \right)$ for all ρ∈S(H). Moreover, $S_{a+b}\left(\rho \right)=S_{a}\left(\rho \right)+S_{b}\left(\rho \right)$ if and only if tr(b)tr(ρa)=tr(a)tr(ρb).*


**Proof.** Since −xlnx is concave, letting $\lambda_{1}=\mathrm{tr}\;\left(a\right)/\left[\mathrm{tr}\;(a)+\mathrm{tr}\;(b)\right]$, $\lambda_{2}=\mathrm{tr}\;\left(b\right)/\left[\mathrm{tr}\;(a)+\mathrm{tr}\;(b)\right]$, $x_{1}=\mathrm{tr}\;\left(\rho a\right)/\mathrm{tr}\;\left(a\right)$, $x_{2}=\mathrm{tr}\;\left(\rho b\right)/\mathrm{tr}\;\left(b\right)$ we obtain
$$\begin{array}{cc} S_{a+b}\left(\rho \right) & =-\mathrm{tr}\;\left[\rho (a+b)\right]\ln\left\{\frac{\mathrm{tr}\;\left[\rho (a+b)\right]}{\mathrm{tr}\;(a+b)}\right\}\\ \; & =-\mathrm{tr}\;\left(a+b\right)\left[\frac{\mathrm{tr}\;(\rho a)+\mathrm{tr}\;(\rho b)}{\mathrm{tr}\;(a+b)}\right]\ln\left[\frac{\mathrm{tr}\;(\rho a)+\mathrm{tr}\;(\rho b)}{\mathrm{tr}\;(a+b)}\right]\\ \; & =-\mathrm{tr}\;\left(a+b\right)\left(\lambda_{1}x_{1}+\lambda_{2}x_{2}\right)\ln\left(\lambda_{1}x_{1}+\lambda_{2}x_{2}\right)\\ \; & \geq -\mathrm{tr}\;\left(a+b\right)\left[\lambda_{1}x_{1}\ln\left(x_{1}\right)+\lambda_{2}x_{2}\ln\left(x_{2}\right)\right]\\ \; & =-\mathrm{tr}\;\left(\rho a\right)\ln\left[\frac{\mathrm{tr}\;(\rho a)}{\mathrm{tr}\;(a)}\right]-\mathrm{tr}\;\left(\rho b\right)\ln\left[\frac{\mathrm{tr}\;(\rho b)}{\mathrm{tr}\;(b)}\right]=S_{a}\left(\rho \right)+S_{b}\left(\rho \right) \end{array}$$
We have equality if and only if $x_{1}=x_{2}$ which is equivalent to tr(b)tr(ρa)=tr(a)tr(ρb). □

**Corollary 1.** 
*$S_{a}\left(\rho \right)+S_{a^{\prime }}\left(\rho \right)\leq \ln\left(n\right)$ and $S_{a}\left(\rho \right)+S_{a^{\prime }}\left(\rho \right)=\ln\left(n\right)$ if and only if tr(a)=ntr(ρa).*


**Proof.** Applying Theorem 2 we obtain
$$S_{a}\left(\rho \right)+S_{a^{\prime }}\left(\rho \right)\leq S_{a+a^{\prime }}\left(\rho \right)=S_{I}\left(\rho \right)=\ln\left(n\right)$$
$$\begin{array}{cc} \mathrm{We}\;\mathrm{have}\;\mathrm{equality}\; & \Leftrightarrow \mathrm{tr}\;\left(a^{\prime }\right)\mathrm{tr}\;\left(\rho a\right)=\mathrm{tr}\;\left(a\right)\mathrm{tr}\;\left(\rho a^{\prime }\right)\\ \; & \Leftrightarrow \left[n-\mathrm{tr}\;(a)\right]\mathrm{tr}\;\left(\rho a\right)=\mathrm{tr}\;\left(a\right)\left[1-\mathrm{tr}\;(\rho a)\right]\\ \; & \Leftrightarrow \mathrm{tr}\;(a)=n\mathrm{tr}\;(\rho a)e \end{array}$$ □

**Corollary 2.** 
*$S_{a+b}\left(\rho \right)\geq S_{a}\left(\rho \right),S_{b}\left(\rho \right)$.*


**Corollary 3.** 
*If a≤b, then $S_{a}\left(\rho \right)\leq S_{b}\left(\rho \right)$ for all ρ∈S(H).*


**Proof.** If a≤b, then b=a+c for c=b−a∈E(H). Hence,
$$S_{b}\left(\rho \right)=S_{a+c}\left(\rho \right)\geq S_{a}\left(\rho \right)+S_{c}\left(\rho \right)\geq S_{a}\left(\rho \right)$$
for every ρ∈S(H). □

Applying Theorem 2 and induction we obtain the following.

**Corollary 4.** 
*If $a_{1}+a_{2}+\cdots +a_{m}\leq I$, then $S_{\sum a_{i}}\left(\rho \right)\geq \sum S_{a_{i}}\left(\rho \right)$. Moreover, we have equality if and only if $\mathrm{tr}\;\left(a_{j}\right)\mathrm{tr}\;\left(\rho a_{i}\right)=\mathrm{tr}\;\left(a_{i}\right)\mathrm{tr}\;\left(\rho a_{j}\right)$ for all i,j=1,2,…,m.*


Notice that E(H) is a convex set in the sense that if $a_{i}\in E\left(H\right)$ and $0\leq \lambda_{i}\leq 1$ with $\sum_{i=1}^{m}\lambda_{i}=1$, then $\sum \lambda_{i}a_{i}\in E\left(H\right)$.

**Corollary 5.** (i) *If 0<λ≤1 and a∈E(H), then $S_{\lambda a}\left(\rho \right)=\lambda S_{a}\left(\rho \right)$ for all ρ∈S(H).* (ii)* If $0<\lambda_{i}\leq 1$, $a_{i}\in E\left(H\right)$, with $\sum_{i=1}^{m}\lambda_{i}=1$, then $S_{\sum \lambda_{i}a_{i}}\left(\rho \right)\leq \sum \lambda_{i}S_{a_{i}}\left(\rho \right)$ for all ρ∈S(H). We have equality if and only if $\mathrm{tr}\;\left(a_{j}\right)\mathrm{tr}\;\left(\rho a_{i}\right)=\mathrm{tr}\;\left(a_{i}\right)\mathrm{tr}\;\left(\rho a_{j}\right)$ for all i,j=1,2,…,m.*

**Proof.** (i) We have that
$$S_{\lambda a}\left(\rho \right)=-\mathrm{tr}\;\left(\rho \lambda a\right)\ln\left[\frac{\mathrm{tr}\;(\rho \lambda a)}{\mathrm{tr}\;(\lambda a)}\right]=-\mathrm{tr}\;\left(\rho a\right)\ln\left[\frac{\lambda \mathrm{tr}\;(\rho a)}{\lambda \mathrm{tr}\;(a)}\right]=\lambda S_{a}\left(\rho \right)$$
(ii) Applying (i) and Corollary 4 gives
$$S_{\sum \lambda_{i}a_{i}}\left(\rho \right)\geq \sum S_{\lambda_{i}a_{i}}\left(\rho \right)=\sum \lambda_{i}S_{a_{i}}\left(\rho \right)$$
together with the equality condition. □

As with E(H), S(H) is a convex set and we have the following.

**Theorem 3.** 
*If $0<\lambda_{i}\leq 1$ $\rho_{i}\in S\left(H\right)$, i=1,2,…,m, with $\sum_{i=1}^{m}\lambda_{i}=1$, then*

$$S_{a}\left(\sum \lambda_{i}\rho_{i}\right)\geq \sum \lambda_{i}S_{a}\left(\rho_{i}\right)$$

*for all a∈E(H). We have equality if and only if $\mathrm{tr}\;\left(\rho_{i}a\right)=\mathrm{tr}\;\left(\rho_{j}a\right)$ for all i,j=1,2,…,m.*


**Proof.** Letting $x_{i}=\mathrm{tr}\;\left(\rho_{i}a\right)/\mathrm{tr}\;\left(a\right)$, since −xlnx is concave, we obtain
$$\begin{array}{cc} S_{a}\left(\sum \lambda_{i}\rho_{i}\right)\; & =-\mathrm{tr}\;\left(\sum \lambda_{i}\rho_{i}a\right)\ln\left[\frac{\mathrm{tr}\;\left(\sum \lambda_{i}\rho_{i}a\right)}{\mathrm{tr}\;(a)}\right]\\ \; & =-\mathrm{tr}\;\left(a\right)\sum \lambda_{i}\frac{\mathrm{tr}\;(\rho_{i}a)}{\mathrm{tr}\;(a)}\ln\left[\frac{\sum \lambda_{i}\mathrm{tr}\;\left(\rho_{i}a\right)}{\mathrm{tr}\;(a)}\right]\\ \; & =\mathrm{tr}\;\left(a\right)\left[-\sum \lambda_{i}x_{i}\ln\left(\sum \lambda_{j}x_{j}\right)\right]\geq -\mathrm{tr}\;\left(a\right)\sum \lambda_{i}x_{i}\ln\left(x_{i}\right)\\ \; & =-\mathrm{tr}\;\left(a\right)\sum \lambda_{i}\frac{\mathrm{tr}\;(\rho_{i}a)}{\mathrm{tr}\;(a)}\ln\left[\frac{\mathrm{tr}\;(\rho_{i}a)}{\mathrm{tr}\;(a)}\right]=-\sum \lambda_{i}\mathrm{tr}\;\left(\rho_{i}a\right)\ln\left[\frac{\mathrm{tr}\;(\rho_{i}a)}{\mathrm{tr}\;(a)}\right]\\ \; & =\sum \lambda_{i}S_{a}\left(\rho_{i}\right) \end{array}$$We have equality if and only if $x_{i}=x_{j}$ which is equivalent to $\mathrm{tr}\;\left(\rho_{i}a\right)=\mathrm{tr}\;\left(\rho_{j}a\right)$ for all i,j=1,2,…,m. □

**Theorem 4.** 
*If $a_{i}\in E\left(H_{i}\right)$, $\rho_{i}\in S\left(H_{i}\right)$, i=1,2, then*

$$S_{a_{1}⊗a_{2}}\left(\rho_{1}⊗\rho_{2}\right)=\mathrm{tr}\;\left(\rho_{2}a_{2}\right)S_{a_{1}}\left(\rho_{1}\right)+\mathrm{tr}\;\left(\rho_{1}a_{1}\right)S_{a_{2}}\left(\rho_{2}\right)\leq S_{a_{1}}\left(\rho_{1}\right)+S_{a_{2}}\left(\rho_{2}\right).$$



**Proof.** This follows from
$$\begin{array}{cc} S_{a_{1}⊗a_{2}}\left(\rho_{1}⊗\rho_{2}\right)\; & =-\mathrm{tr}\;\left(\rho_{1}⊗\rho_{2}a_{1}⊗a_{2}\right)\ln\left[\frac{\mathrm{tr}\;(\rho_{1}⊗\rho_{2}a_{1}⊗a_{2})}{\mathrm{tr}\;(a_{1}⊗a_{2})}\right]\\ \; & =-\mathrm{tr}\;\left(\rho_{1}a_{1}\right)\mathrm{tr}\;\left(\rho_{2}a_{2}\right)\ln\left[\frac{\mathrm{tr}\;\left(\rho_{1}a_{1}\right)\mathrm{tr}\;\left(\rho_{2}a_{2}\right)}{\mathrm{tr}\;\left(a_{1}\right)\mathrm{tr}\;\left(a_{2}\right)}\right]\\ \; & =-\mathrm{tr}\;\left(\rho_{1}a_{1}\right)\mathrm{tr}\;\left(\rho_{2}a_{2}\right)\left\{\ln\left[\frac{\mathrm{tr}\;(\rho_{1}a_{1})}{\mathrm{tr}\;(a_{1})}\right]+\ln\left[\frac{\mathrm{tr}\;(\rho_{2}a_{2})}{\mathrm{tr}\;(a_{2})}\right]\right\}\\ \; & =\mathrm{tr}\;\left(\rho_{2}a_{2}\right)S_{a_{1}}\left(\rho_{1}\right)+\mathrm{tr}\;\left(\rho_{1}a_{1}\right)S_{a_{2}}\left(\rho_{2}\right)\leq S_{a_{1}}\left(\rho_{1}\right)+S_{a_{2}}\left(\rho_{2}\right)e \end{array}$$ □

An *operation* on *H* is a completely positive linear map I:L(H)→L(H) such that $\mathrm{tr}\;\left[I(A)\right]\leq \mathrm{tr}\;\left(A\right)$ for all A∈L(H) [2,3,6,9,10]. If I is an operation we define the *dual* of I to be the unique linear map $I^{*}:L\left(H\right)\to L\left(H\right)$ that satisfies $\mathrm{tr}\;\left[I(A)B\right]=\mathrm{tr}\;\left[AI^{*}\left(B\right)\right]$ for all A,B∈L(H). If a∈E(H) then for any ρ∈S(H) we have $0\leq \mathrm{tr}\;\left[I(\rho )a\right]\leq 1$ and it follows that $I^{*}\left(a\right)\in E\left(H\right)$. We say that I*measures*a∈E(H) if $\mathrm{tr}\;\left[I(\rho )\right]=\mathrm{tr}\;\left(\rho a\right)$ for all ρ∈S(H). If I measures *a* we define the I-*sequential product*$a\circ b=I^{*}\left(b\right)$ for all b∈E(H) [12,13]. Although a∘b depends on the operation used to measure *a* we do not include I in the notation for simplicity. We interpret a∘b as the effect that results from first measuring *a* using I and then measuring *b*.

**Theorem 5.** (i) *If b⊥c, then a∘(b+c)=a∘b+a∘c.* (ii)*a∘I=a. *(iii)*a∘b≤a for all b∈E(H).* (iv)*$S_{a\circ b}\left(\rho \right)\leq S_{a}\left(\rho \right)$ for all ρ∈S(H).*

**Proof.** (i) For every ρ∈S(H) we obtain
$$\begin{array}{cc} \mathrm{tr}\;\left[\rho \;a\circ (b+c)\right] & =\mathrm{tr}\;\left[\rho I^{*}\left(b+c\right)\right]=\mathrm{tr}\;\left[I(\rho )(b+c)\right]=\mathrm{tr}\;\left[I(\rho )b\right]+\mathrm{tr}\;\left[I(\rho )c\right]\\ \; & =\mathrm{tr}\;\left[\rho I^{*}\left(b\right)\right]+\mathrm{tr}\;\left[\rho I^{*}\left(c\right)\right]=\mathrm{tr}\;\left[\rho \;a\circ b\right]+\mathrm{tr}\;\left[\rho \;a\circ c\right]\\ \; & =\mathrm{tr}\;\left[\rho (a\circ b+a\circ c)\right] \end{array}$$
Hence, a∘(b+c)=a∘b+a∘c. (ii) For all ρ∈S(H) we have
$$\mathrm{tr}\;\left(\rho \;a\circ I\right)=\mathrm{tr}\;\left[\rho I^{*}\left(I\right)\right]=\mathrm{tr}\;\left[I(\rho )I\right]=\mathrm{tr}\;\left[I(\rho )\right]=\mathrm{tr}\;\left(\rho a\right)$$
Hence, a∘I=a. (iii) By (i) and (ii) we have
$$a\circ b+a\circ b^{\prime }=a\circ \left(b+b^{\prime }\right)=a\circ I=a$$
It follows that a∘b≤a. (iv) Since a∘b≤a, by Corollary 3 we obtain $S_{a\circ b}\left(\rho \right)\leq S_{a}\left(\rho \right)$ for all ρ∈S(H). □

Theorem 5(iv) shows that a∘b gives more information than *a* about ρ. We can continue this process and make more measurements as follows. If $I^{i}$ measures $a^{i}$, i=1,2,…,m, we have
$$a^{1}\circ a^{2}\circ \cdots \circ a^{m}={\left(I^{1}\right)}^{*}{\left(I^{2}\right)}^{*}\cdots {\left(I^{m-1}\right)}^{*}\left(a^{m}\right)$$
and it follows from Theorem 5(iv) that
$$S_{a^{1}\circ a^{2}\circ \cdots \circ a^{m}}\left(\rho \right)\leq S_{a^{1}\circ a^{2}\circ \cdots \circ a^{m-1}}\left(\rho \right)$$
Notice that the probability of occurrence of the effect $a^{1}\circ a^{2}\circ \cdot \circ a^{m}$ in state ρ is
$$\begin{array}{cc} \mathrm{tr}\;(\rho \;a^{1}\circ a^{2}\circ \cdots \circ a^{m}) & =\mathrm{tr}\;\left[\rho {\left(I^{1}\right)}^{*}{\left(I^{2}\right)}^{*}\cdots {\left(I^{m-1}\right)}^{*}\left(a^{m}\right)\right]\\ \; & =\mathrm{tr}\;\left[I^{m-1}I^{m-2}\cdots I^{1}\left(\rho \right)a^{m}\right] \end{array}$$
Thus, we begin with the input state ρ, then measure $a^{1}$ using $I^{1}$, then measure $a^{2}$ using $I^{2},\ldots$ and finally measuring $a^{m}$.

**Example 1.** 
*1 For a∈E(H) we define the Lüders operation $L^{a}\left(A\right)=a^{1/2}Aa^{1/2}$ [14]. Since*

$$\mathrm{tr}\;\left[A{\left(L^{a}\right)}^{*}\left(B\right)\right]=\left[L^{a}\left(A\right)B\right]=\mathrm{tr}\;\left[a^{1/2}Aa^{1/2}B\right]=\mathrm{tr}\;\left(Aa^{1/2}Ba^{1/2}\right)$$

*we have ${\left(L^{a}\right)}^{*}\left(B\right)=a^{1/2}Ba^{1/2}$ so ${\left(L^{a}\right)}^{*}=L^{a}$. We have that $L^{a}$ measures a because*

$$\mathrm{tr}\;\left[L^{a}\left(\rho \right)\right]=\mathrm{tr}\;\left(a^{1/2}\rho a^{1/2}\right)=\mathrm{tr}\;\left(\rho a\right)$$

*for every ρ∈S(H). We conclude that the $L^{a}$ sequential product is*

$$a\circ b={\left(L^{a}\right)}^{*}\left(b\right)=a^{1/2}ba^{1/2}$$

*We also have that*

$$\begin{array}{cc} S_{a\circ b}\left(\rho \right) & =-\mathrm{tr}\;\left(\rho \;a\circ b\right)\ln\left[\frac{\mathrm{tr}\;(\rho \;a\circ b)}{\mathrm{tr}\;(a\circ b)}\right]=-\mathrm{tr}\;\left(\rho \;a^{1/2}ba^{1/2}\right)\ln\left[\frac{\mathrm{tr}\;(\rho \;a^{1/2}ba^{1/2})}{\mathrm{tr}\;(a^{1/2}ba^{1/2})}\right]\\ \; & =-\mathrm{tr}\;\left(a\circ \rho \;b\right)\ln\left[\frac{\mathrm{tr}\;(a\circ \rho \;b)}{\mathrm{tr}\;(ab)}\right].\; \end{array}$$



**Example 2.** 
*2 For a∈E(H), α∈S(H) we define the Holevo operation [15] $H^{\left(a,\alpha \right)}\left(A\right)=\mathrm{tr}\;\left(Aa\right)\alpha$. Since*

$$\begin{array}{cc} \mathrm{tr}\;\left[A{\left(H^{\left(a,\alpha \right)}\right)}^{*}\left(B\right)\right] & =\mathrm{tr}\;\left[H^{\left(a,\alpha \right)}\left(A\right)B\right]=\mathrm{tr}\;\left[\mathrm{tr}\;(Aa)\alpha B\right]=\mathrm{tr}\;\left(Aa\right)\mathrm{tr}\;\left(\alpha B\right)\\ \; & =\mathrm{tr}\;\left[A\mathrm{tr}\;(\alpha B)a\right] \end{array}$$

*we have ${\left(H^{\left(a,\alpha \right)}\right)}^{*}\left(B\right)=\mathrm{tr}\;\left(\alpha B\right)a$. We have $H^{\left(a,\alpha \right)}$ measures a because*

$$\mathrm{tr}\;\left[H^{\left(a,\alpha \right)}\left(\rho \right)\right]=\mathrm{tr}\;\left(\rho a\right)$$

*for every ρ∈S(H). We conclude that the $H^{\left(a,\alpha \right)}$ sequential product is*

$$a\circ b={\left(H^{\left(a,\alpha \right)}\right)}^{*}\left(b\right)=\mathrm{tr}\;\left(\alpha b\right)a$$

*We also have that*

$$S_{a\circ b}\left(\rho \right)=-\mathrm{tr}\;\left(\alpha b\right)\mathrm{tr}\;\left(\rho a\right)\ln\left[\frac{\mathrm{tr}\;(\rho a)}{\mathrm{tr}\;(a)}\right]=\mathrm{tr}\;\left(\alpha b\right)S_{a}\left(\rho \right)$$

*If $a_{i}\in E\left(H\right)$, i=1,2,…,m, and we measure $a_{i}$ with operations $H^{\left(a_{i},\alpha_{i}\right)}$, i=1,2,…,m−1, then*

$$\begin{array}{cc} a_{1}\circ a_{2}\circ \cdots \circ a_{m} & =a_{1}\circ \left(a_{2}\circ \cdots \circ a_{m}\right)=\mathrm{tr}\;\left(\alpha_{1}a_{2}\circ \cdots \circ a_{m}\right)a_{1}\\ \; & =\mathrm{tr}\;\left[\alpha_{1}\mathrm{tr}\;\left(\alpha_{2}a_{3}\circ \cdots \circ a_{m}\right)a_{2}\right]a_{1}\\ \; & =\mathrm{tr}\;\left(\alpha_{2}a_{3}\circ \cdots \circ a_{m}\right)\mathrm{tr}\;\left(\alpha_{1}a_{2}\right)a_{1}\\ \; & \vdots \\ \; & =\mathrm{tr}\;\left(\alpha_{m-1}a_{m}\right)\mathrm{tr}\;\left(\alpha_{m-2}a_{m-1}\right)\cdots \mathrm{tr}\;\left(\alpha_{1}a_{2}\right)a_{1} \end{array}$$

*Moreover, it follows from Corollary 5(i) that*

$$S_{a_{1}\circ \cdots \circ a_{m}}\left(\rho \right)=\mathrm{tr}\;\left(\alpha_{m-1}a_{m}\right)\mathrm{tr}\;\left(\alpha_{m-2}a_{m-1}\right)\cdots \mathrm{tr}\;\left(\alpha_{1}a_{2}\right)S_{a_{1}}\left(\rho \right)$$

*for all ρ∈S(H).*


## 3. Entropy of Observables and Instruments

We now extend our work on entropy of effects to entropy of observables and instruments. An *observable* on *H* is a finite collection of effects $A=\left\{A_{x}:x\in \Omega_{A}\right\}$, $A_{x}\neq 0$, where $\sum_{x\in \Omega_{A}}A_{x}=I$ [2,3,9]. The set $\Omega_{A}$ is called the *outcome space* of *A*. The effect $A_{x}$ occurs when a measurement of *A* results in the outcome *x*. If ρ∈S(H), then $\mathrm{tr}\;(\rho A_{x})$ is the probability that outcome *x* results from a measurement of *A* when the system is in state ρ. If $\Delta \subseteq \Omega_{A}$, then
$$\Phi_{\rho }^{A}\left(\Delta \right)=\sum_{x\in \Delta }\mathrm{tr}\;\left(\rho A_{x}\right)$$
is the probability that *A* has an outcome in Δ when the system is in state ρ and $\Phi_{\rho }^{A}$ is called the *distribution* of *A*. We also use the notation $A\left(\Delta \right)=\sum \left\{A_{x}:x\in \Delta \right\}$ so $\Phi_{\rho }^{A}\left(\Delta \right)=\mathrm{tr}\;\left[\rho A(\Delta )\right]$ for all $\Delta \subseteq \Omega_{A}$. In this way, an observable is a *positive operation-valued measure* (POVM). We say that an observable *A* is *sharp* if $A_{x}$ is a projection on *H* for all $x\in \Omega_{A}$ and *A* is *atomic* if $A_{x}$ is a one-dimensional projection for all $x\in \Omega_{A}$.

If *A* is an observable and ρ∈S(H) the ρ-*entropy* of *A* is $S_{A}\left(\rho \right)=\sum S_{A_{x}}\left(\rho \right)$ where the sum is over the $x\in \Omega_{A}$ such that $\mathrm{tr}\;(\rho A_{x})\neq 0$. Then $S_{A}\left(\rho \right)$ is a measure of the information that a measurement of *A* gives about ρ. The smaller $S_{A}\left(\rho \right)$ is, the more information given. Notice that if *A* is sharp, then $\mathrm{tr}\;\left(A_{x}\right)=\dim\left(A_{x}\right)$ and if *A* is atomic, then
$$S_{A}\left(\rho \right)=-\sum_{x}\mathrm{tr}\;\left(\rho A_{x}\right)\ln\left[\mathrm{tr}\;(\rho A_{x})\right]$$
There are two interesting extremes for $S_{A}\left(\rho \right)$. If ρ has spectral decomposition $\rho =\sum_{i=1}^{m}\lambda_{i}P_{i}$ and *A* is the observable $A=\left\{P_{i}:i=1,2,\ldots ,m\right\}$, then
$$S_{A}\left(\rho \right)=-\sum_{i}\mathrm{tr}\;\left(\rho P_{i}\right)\ln\left[\mathrm{tr}\;(\rho P_{i})\right]=-\sum \lambda_{i}\ln\left(\lambda_{i}\right)=S\left(\rho \right)$$
As we shall see, this gives the minimum entropy (most information). For the completely random state I/n and any observable *A* we obtain
(3)$$\begin{array}{cc} S_{A}\left(I/n\right) & =-\sum_{x}\frac{\mathrm{tr}\;(A_{x})}{n}\ln\left[\frac{\mathrm{tr}\;(A_{x})/n}{\mathrm{tr}\;(A_{x})}\right]=-\frac{1}{n}\sum_{x}\mathrm{tr}\;\left(A_{x}\right)\ln\left(\frac{1}{n}\right)\\ \; & =\frac{\ln(n)}{n}\sum_{x}\mathrm{tr}\;\left(A_{x}\right)=\frac{\ln(n)}{n}\;\mathrm{tr}\;\left(I\right)=\ln\left(n\right) \end{array}$$
We shall also see that this gives the maximum entropy (least information).

**Theorem 6.** 
*For any observable A and ρ∈S(H) we have*

$$S\left(\rho \right)\leq S_{A}\left(\rho \right)\leq \ln\left(n\right)$$



**Proof.** Applying Theorem 1 we obtain
$$\begin{array}{cc} S_{A}\left(\rho \right) & =\sum_{x\in \Omega_{A}}S_{A_{x}}\left(\rho \right)\geq -\sum_{x\in \Omega_{A}}\sum_{i}\mathrm{tr}\;\left(P_{i}A_{x}\right)\lambda_{i}\ln\left(\lambda_{i}\right)\\ \; & =-\sum_{i}\mathrm{tr}\;\left(P_{i}\sum_{x\in \Omega_{A}}A_{x}\right)\lambda_{i}\ln\left(\lambda_{i}\right)\\ \; & =-\sum_{i}\mathrm{tr}\;\left(P_{i}\right)\lambda_{i}\ln\left(\lambda_{i}\right)=-\sum_{i}\lambda_{i}\ln\left(\lambda_{i}\right)=S\left(\rho \right) \end{array}$$
Since ln(x) is concave and $\mathrm{tr}\;(\rho A_{x})>0$, $\sum_{x}\mathrm{tr}\;\left(\rho A_{x}\right)=1$ we have by Jensen’s inequality
$$\begin{array}{cc} S_{A}\left(\rho \right) & =\sum_{x}\mathrm{tr}\;\left(\rho A_{x}\right)\ln\left[\frac{\mathrm{tr}\;(A_{x})}{\mathrm{tr}\;(\rho A_{x})}\right]\leq \ln\left[\sum_{x}\mathrm{tr}\;\left(\rho A_{x}\right)\frac{\mathrm{tr}\;(A_{x})}{\mathrm{tr}\;(\rho A_{x})}\right]\\ \; & =\ln\left[\sum_{x}\mathrm{tr}\;\left(A_{x}\right)\right]=\ln\left[\mathrm{tr}\;(I)\right]=\ln\left(n\right)e \end{array}$$ □

An observable *A* is *trivial* if $A_{x}=\lambda_{x}I$, $0<\lambda_{x}\leq 1$, $\sum \lambda_{x}=1$.

**Corollary 6.** (i) *$S_{A}\left(\rho \right)=\ln\left(n\right)$ if and only if $\mathrm{tr}\;\left(A_{x}\right)\mathrm{tr}\;\left(\rho A_{y}\right)=\mathrm{tr}\;\left(A_{y}\right)\mathrm{tr}\;\left(\rho A_{x}\right)$ for all $x,y\in \Omega_{A}$.* (ii)* A is trivial if and only if $S_{A}\left(\rho \right)=\ln\left(n\right)$ for all ρ∈S(H).* (iii)*ρ=I/n if and only if $S_{A}\left(\rho \right)=\ln\left(n\right)$ for all observables A. *(iv)*S(ρ)=ln(n) if and only if ρ=I/n.*

**Proof.** (i) This follows from the proof of Theorem 6 because this is the condition for equality in Jensen’s inequality. (ii) Suppose *A* is trivial with $A_{x}=\lambda_{x}I$. Then for every ρ∈S(H) we have
$$S_{A}\left(\rho \right)=-\sum_{x}\mathrm{tr}\;\left(\rho \lambda_{x}I\right)\ln\left[\frac{\mathrm{tr}\;(\rho \lambda_{x}I)}{\mathrm{tr}\;(\lambda_{x}I)}\right]=-\sum_{x}\lambda_{x}\ln\left(\frac{\lambda_{x}}{n\lambda_{x}}\right)=\ln\left(n\right)\sum_{x}\lambda_{x}=\ln\left(n\right)$$
Conversely, suppose $S_{A}\left(\rho \right)=\ln\left(n\right)$ for all ρ∈S(H). By (i) we have that $\mathrm{tr}\;\left(A_{x}\right)\mathrm{tr}\;\left(\rho A_{y}\right)=\mathrm{tr}\;\left(A_{y}\right)\mathrm{tr}\;\left(\rho A_{x}\right)$ for all ρ∈S(H). It follows that
$$\langle \phi ,A_{y}\phi \rangle =\langle \phi ,A_{x}\phi \rangle \frac{\mathrm{tr}\;(A_{y})}{\mathrm{tr}\;(A_{x})}$$
for every ϕ∈H, ϕ≠0. Hence, $A_{y}=\left(\mathrm{tr}\;\left(A_{y}\right)\right)/\left(\mathrm{tr}\;\left(A_{x}\right)\right)A_{x}$ so that
$$I=\sum_{y}A_{y}=\sum_{y}\frac{\mathrm{tr}\;(A_{y})}{\mathrm{tr}\;(A_{x})}\;A_{x}=\frac{n}{\mathrm{tr}\;(A_{x})}\;A_{x}$$
We conclude that $A_{x}=\left(\mathrm{tr}\;\left(A_{x}\right)\right)/n\;I$ for all $x\in \Omega_{A}$ so *A* is trivial. (iii) If ρ=I/n, we have shown in (Equation 3) that $S_{A}\left(\rho \right)=\ln\left(n\right)$ for all observables *A*. Conversely, if $S_{A}\left(\rho \right)=\ln\left(n\right)$ for every observable *A*, as before, we have $\mathrm{tr}\;\left(A_{x}\right)\mathrm{tr}\;\left(\rho A_{y}\right)=\mathrm{tr}\;\left(A_{y}\right)\mathrm{tr}\;\left(\rho A_{x}\right)$ for every observable *A*. Letting $A_{x}$ be the observable given by the spectral decomposition $\rho =\sum \lambda_{x}A_{x}$ where *A* is atomic, we conclude that $\lambda_{x}=\lambda_{y}$ for all $x,y\in \Omega_{A}$. Hence, $\lambda_{x}=1/n$ and $\rho =\sum \left(1/n\right)A_{x}=I/n$. (iv)If S(ρ)=ln(n), by Theorem 6, $S_{A}\left(\rho \right)=\ln\left(n\right)$ for every observable *A*. Applying (iii), ρ=I/n. Conversely, if ρ=I/n, then
$$S\left(\rho \right)=-\sum_{i=1}^{n}\frac{1}{n}\;\ln\left(\frac{1}{n}\right)=-\ln\left(\frac{1}{n}\right)=\ln\left(n\right)e$$ □

We now extend Corollary 5(ii) and Theorem 3 to observables. If $A^{i}=\left\{A_{x}^{i}:x\in \Omega \right\}$ are observables with the same outcome space Ω, i=1,2,…,m, and $0<\lambda_{i}\leq 1$ with $\sum_{i=1}^{m}\lambda_{i}=1$, then the observable $A=\left\{A_{x}:x\in \Omega \right\}$ where $A_{x}=\sum_{i=1}^{m}\lambda_{i}A_{x}^{i}$ is called a *convex combination* of the $A^{i}$ [12].

**Theorem 7.** (i) *If A is a convex combination of $A^{i}$, i=1,2,…,m, then for all ρ∈S(H) we have*
$$S_{A}\left(\rho \right)\geq \sum_{i=1}^{m}\lambda_{i}S_{A^{i}}\left(\rho \right)$$
(ii) *If $0<\lambda_{i}\leq 1$ with $\sum_{i=1}^{m}\lambda_{i}=1$, $\rho_{i}\in S\left(H\right)$, i=1,2,…,m, and A is an observable, then*
$$S_{A}\left(\sum_{i}\lambda_{i}\rho_{i}\right)\geq \sum_{i}\lambda_{i}S_{A}\left(\rho_{i}\right)$$

**Proof.** (i) Applying Corollary 5(ii) gives
$$\begin{array}{cc} S_{A}\left(\rho \right) & =\sum_{x}S_{A_{x}}\left(\rho \right)=\sum_{x}S_{\sum \lambda_{i}A_{x}^{i}}\left(\rho \right)\geq \sum_{x}\sum_{i}\lambda_{i}S_{A_{x}^{i}}\left(\rho \right)\\ \; & =\sum_{i}\lambda_{i}\sum_{x}S_{A_{x}^{i}}\left(\rho \right)=\sum_{i}\lambda_{i}S_{A^{i}}\left(\rho \right) \end{array}$$
(ii) Applying Theorem 3 gives
$$\begin{array}{cc} S_{A}\left(\sum_{i}\lambda_{i}\rho_{i}\right) & =\sum_{x}S_{A_{x}}\left(\sum_{i}\lambda_{i}\rho_{i}\right)\geq \sum_{x}\sum_{i}\lambda_{i}S_{A_{x}}\left(\rho_{i}\right)\\ \; & =\sum_{i}\lambda_{i}\sum_{x}S_{A_{x}}\left(\rho_{i}\right)=\sum_{i}\lambda_{i}S_{A}\left(\rho_{i}\right)e \end{array}$$ □

We say that an observable *B* is a *coarse-graining* of an observable *A* if there exists a surjection $f:\Omega_{A}\to \Omega_{B}$ such that
$$B_{y}=\sum \left\{A_{x}:f\left(x\right)=y\right\}=A\left[f^{-1}\left(y\right)\right]$$
for every $y\in \Omega_{B}$ [2,12,16].

**Theorem 8.** 
*If B is a coarse-graining of A, then $S_{B}\left(\rho \right)\geq S_{A}\left(\rho \right)$ for all ρ∈S(H).*


**Proof.** Let $B_{y}=A\left[f^{-1}\left(y\right)\right]$ for all $y\in \Omega_{B}$ and let $p_{y}=\mathrm{tr}\;\left(\rho B_{y}\right)$, $p_{x}^{\prime }=\mathrm{tr}\;\left(\rho A_{x}\right)$ for all $y\in \Omega_{b}$, $x\in \Omega_{A}$. Then
$$p_{y}=\mathrm{tr}\;\left(\rho \sum_{f(x)=y}A_{x}\right)=\sum_{f(x)=y}\mathrm{tr}\;\left(\rho A_{x}\right)=\sum_{f(x)=y}p_{x}^{\prime }$$
Let $V_{y}=\mathrm{tr}\;\left(B_{y}\right)$, $V_{x}^{\prime }=\mathrm{tr}\;\left(A_{x}\right)$ so that
$$V_{y}=\mathrm{tr}\;\sum \left(\sum_{f(x)=y}A_{x}\right)=\sum_{f(x)=y}\mathrm{tr}\;\left(A_{x}\right)=\sum_{f(x)=y}V_{x}^{\prime }$$
Since −xln(x) is concave, we conclude that
$$\begin{array}{cc} S_{B}\left(\rho \right) & =-\sum_{y}p_{y}\ln\left(\frac{p_{y}}{V_{y}}\right)=-\sum_{y}\sum_{f(x)=y}p_{x}^{\prime }\ln\left[\frac{\sum_{f(x)=y}p_{x}^{\prime }}{V_{y}}\right]\\ \; & =-\sum_{y}V_{y}\left(\sum_{f(x)=y}\frac{p_{x}^{\prime }V_{x}^{\prime }}{V_{x}^{\prime }V_{y}}\right)\ln\left(\sum_{f(x)=y}\frac{p_{x}^{\prime }V_{x}^{\prime }}{V_{x}^{\prime }V_{y}}\right)\\ \; & \geq -\sum_{y}V_{y}\sum_{f(x)=y}\frac{V_{x}^{\prime }}{V_{y}}\left[\frac{p_{x}^{\prime }}{V_{x}^{\prime }}\;\ln\left(\frac{p_{x}^{\prime }}{V_{x}^{\prime }}\right)\right]=-\sum_{y}\sum_{f(x)=y}p_{x}^{\prime }\ln\left(\frac{p_{x}^{\prime }}{V_{x}^{\prime }}\right)\\ \; & =-\sum_{x}p_{x}^{\prime }\ln\left(\frac{p_{x}^{\prime }}{V_{x}^{\prime }}\right)=S_{A}\left(\rho \right)e \end{array}$$ □

The equality condition for Jensen’s inequality gives the following.

**Corollary 7.** 
*An observable A possesses a coarse-graining $B_{y}=A\left[f^{-1}\left(y\right)\right]$ with $S_{B}\left(\rho \right)=S_{A}\left(\rho \right)$ for all ρ∈S(H) if and only if for every $x_{1},x_{2}\in \Omega_{A}$ with $f\left(x_{1}\right)=f\left(x_{2}\right)$ we have*

$$\mathrm{tr}\;\left(A_{x_{2}}\right)\mathrm{tr}\;\left(\rho A_{x_{1}}\right)=\mathrm{tr}\;\left(A_{x_{1}}\right)\mathrm{tr}\;\left(\rho A_{x_{2}}\right)$$



A trace preserving operation is called a *channel*. An *instrument* on *H* is a finite collection of operations $I=\left\{I_{x}:x\in \Omega \right\}$ such that $\sum_{x\in \Omega_{I}}I_{x}$ is a channel [2,3,9]. We call $\Omega_{I}$ the *outcome space* for I. If I is an instrument, there exists a unique observable *A* such that $\mathrm{tr}\;\left(\rho A_{x}\right)=\mathrm{tr}\;\left[I_{x}\left(\rho \right)\right]$ for all $x\in \Omega_{A}=\Omega_{I}$, ρ∈S(H) and we say that I *measures A*. Although an instrument measures a unique observable, an observable is measured by many instruments For example, if *A* is an observable, the corresponding *Łüders instrument* [14] is defined by
$$L_{x}^{A}\left(B\right)=A_{x}^{1/2}BA_{x}^{1/2}$$
for all B∈L(H). Then $L^{A}$ is an instrument because
$$\begin{array}{cc} \mathrm{tr}\;\left[\sum_{x}L_{x}^{A}\left(B\right)\right] & =\sum_{x}\mathrm{tr}\;\left[L_{x}^{A}\left(B\right)\right]=\sum_{x}\mathrm{tr}\;\left(A_{x}^{1/2}BA_{x}^{1/2}\right)=\sum_{x}\mathrm{tr}\;\left(A_{x}B\right)\\ \; & =\mathrm{tr}\;\left(\sum_{x}A_{x}B\right)=\mathrm{tr}\;\left(IB\right)=\mathrm{tr}\;\left(B\right) \end{array}$$
for all B∈L(H). Moreover, $L^{A}$ measures *A* because
$$\mathrm{tr}\;\left[L_{x}^{A}\left(\rho \right)\right]=\mathrm{tr}\;\left(A_{x}^{1/2}\rho A_{x}^{1/2}\right)=\mathrm{tr}\;\left(\rho A_{x}\right)$$
for all ρ∈S(H). Of course, this is related to Example 1. Corresponding to Example 2, we have a *Holevo instrument* $H^{\left(A,\alpha \right)}$ where $\alpha_{x}\in S\left(H\right)$, $x\in \Omega_{A}$ and
$$H_{x}^{\left(A,\alpha \right)}\left(B\right)=\mathrm{tr}\;\left(BA_{x}\right)\alpha_{x}$$
for all B∈L(H) [15]. To show that $H^{\left(A,\alpha \right)}$ is an instrument we have
$$\begin{array}{cc} \mathrm{tr}\;\left[\sum_{x}H_{x}^{\left(A,\alpha \right)}\left(B\right)\right] & =\sum_{x}\mathrm{tr}\;\left[H_{x}^{\left(A,\alpha \right)}\left(B\right)\right]=\sum_{x}\mathrm{tr}\;\left[\mathrm{tr}\;\left(BA_{x}\right)\alpha_{x}\right]\\ \; & =\sum_{x}\mathrm{tr}\;\left(BA_{x}\right)=\mathrm{tr}\;\left(B\sum_{x}A_{x}\right)=\mathrm{tr}\;\left(B\right) \end{array}$$
Moreover, $H^{\left(A,\alpha \right)}$ measures *A* because
$$\mathrm{tr}\;\left[H_{x}^{A,\alpha }\left(\rho \right)\right]=\mathrm{tr}\;\left[\left(\rho A_{x}\right)\alpha_{x}\right]=\mathrm{tr}\;\left(\rho A_{x}\right)\mathrm{tr}\;\left(\alpha_{x}\right)=\mathrm{tr}\;\left(\rho A_{x}\right)$$

Let A,B be observables and let I be an instrument that measures *A*. We define the I-*sequential* product A∘B [12,13] by $\Omega_{A\circ B}=\Omega_{A}\times \Omega_{B}$ and
$$A\circ B_{\left(x,y\right)}=I_{x}^{*}\left(B_{y}\right)=A_{x}\circ B_{y}$$
Defining $f:\Omega_{A\circ B}\to \Omega_{A}$ by f(x,y)=x,we obtain
$$A\circ B\left[f^{-1}\left(x\right)\right]=\sum_{f(x,y)=x}A_{x}\circ B_{y}=\sum_{y\in \Omega_{B}}I_{x}^{*}\left(B_{y}\right)=I_{\alpha }^{*}\left(I\right)=A_{x}$$
We conclude that *A* is a coarse-graining of A∘B. Applying Theorem 8 we obtain the following.

**Corollary 8.** 
*If A,B are observables, the $S_{A\circ B}\left(\rho \right)\leq S_{A}\left(\rho \right)$ for all ρ∈S(H). Equality $S_{A\circ B}\left(\rho \right)=S_{A}\left(\rho \right)$ holds if and only if for every $x\in \Omega_{A}$, $y_{1},y_{2}\in \Omega_{B}$ we have*

$$\frac{\mathrm{tr}\;(\rho A_{x}\circ B_{y_{1}})}{\mathrm{tr}\;(A_{x}\circ B_{y_{1}})}\;\ln\left[\frac{\mathrm{tr}\;(\rho A_{x}\circ B_{y_{1}})}{\mathrm{tr}\;(A_{x}\circ B_{y_{1}})}\right]=\frac{\mathrm{tr}\;(\rho A_{x}\circ B_{y_{2}})}{\mathrm{tr}\;(A_{x}\circ B_{y_{2}})}\;\ln\left[\frac{\mathrm{tr}\;(\rho A_{x}\circ B_{y_{2}})}{\mathrm{tr}\;(A_{x}\circ B_{y_{2}})}\right]$$



Extending this work to more than two observables, let $I^{1},I^{2},\ldots ,I^{m-1}$ be instruments that measure the observables $A^{1},A^{2},\ldots ,A^{m-1}$, respectively. If $A^{m}$ is another observable, we have that
$${\left(A^{1}\circ A^{2}\circ \cdots \circ A^{m}\right)}_{\left(x_{1},x_{2},\ldots ,x_{m}\right)}={\left(I_{x_{1}}^{1}\right)}^{*}{\left(I_{x_{2}}^{2}\right)}^{*}\cdots {\left(I_{x_{m-1}}^{m-1}\right)}^{*}\left(A_{x_{m}}^{m}\right)$$

The next result follows from Corollary 8.

**Corollary 9.** 
*If $A^{1},A^{2},\ldots ,A^{m}$ are observables, then*

$$S_{A^{1}\circ A^{2}\circ \cdots \circ A^{m}}\left(\rho \right)\leq S_{A^{1}\circ A^{2}\circ \cdots \circ A^{m-1}}\left(\rho \right)$$

*for all ρ∈S(H).*


If I is an instrument, let *A* be the unique observable that I measures so $\mathrm{tr}\;\left[I_{x}\left(\rho \right)\right]=\mathrm{tr}\;\left(\rho A_{x}\right)$ for all $x\in \Omega_{I}$ and ρ∈S(H). We define the ρ-*entropy* of I as $S_{I}\left(\rho \right)=S_{A}\left(\rho \right)$. Since $A_{x}=I_{x}^{*}\left(I\right)$ we have
$$\mathrm{tr}\;\left(A_{x}\right)=\mathrm{tr}\;\left[I_{x}^{*}\left(I\right)\right]=\mathrm{tr}\;\left[I_{x}\left(I\right)\right]$$
Hence,
$$S_{I}\left(\rho \right)=S_{A}\left(\rho \right)=-\sum_{x}\mathrm{tr}\;\left(\rho A_{x}\right)\ln\left[\frac{\mathrm{tr}\;(\rho A_{x})}{\mathrm{tr}\;(A_{x})}\right]=-\sum_{x}\mathrm{tr}\;\left[I_{x}\left(\rho \right)\right]\ln\left\{\frac{\mathrm{tr}\;\left[I_{x}\left(\rho \right)\right]}{\mathrm{tr}\;\left[I_{x}\left(I\right)\right]}\right\}$$
Now let $I^{1},I^{2},\ldots ,I^{m}$ be instruments and let $A^{1},A^{2},\ldots ,A^{m}$ be the unique observables they measure, respectively. Denoting the composition of two instruments I,J by I∘J we have
$$\begin{array}{cc} \mathrm{tr}\;\left[I_{x_{m}}^{m}\circ I_{x_{m-1}}^{m-1}\circ \cdots \circ I_{x_{1}}^{1}\left(\rho \right)\right]\; & =\mathrm{tr}\;\left[\rho {\left(I_{x_{1}}^{1}\right)}^{*}{\left(I_{x_{2}}^{1}\right)}^{*}\cdots {\left(I_{x_{m}}^{m}\right)}^{*}\left(I\right)\right]\\ \; & =\mathrm{tr}\;(\rho A_{x_{1}}^{1}\circ A_{x_{2}}^{2}\circ \cdots \circ A_{x_{m}}^{m}) \end{array}$$
Hence, the observable measured by $I^{m}\circ I^{m-1}\circ \cdots \circ I^{1}$ is $A^{1}\circ A^{2}\circ \cdots \circ A^{m}$. It follows that
$$S_{I^{m}\circ I^{m-1}\circ \cdots \circ I^{1}}\left(\rho \right)=S_{A^{1}\circ A^{2}\circ \cdots \circ A^{m}}\left(\rho \right)$$
We conclude that Theorems 1, 2 and 3 [1] follow from our results. Moreover, our proofs are simpler since they come from the more basic concept of ρ-entropy for effects.

Let A,B be observables on *H* and let I be an instrument that measures *A*. The corresponding sequential product becomes
$${\left(A\circ B\right)}_{\left(x,y\right)}=I_{x}^{*}\left(B_{y}\right)=A_{x}\circ B_{y}$$
The ρ-entropy of A∘B has the form
$$\begin{array}{cc} S_{A\circ B}\left(\rho \right)\; & =-\sum_{x,y}\mathrm{tr}\;\left[\rho {\left(A\circ B\right)}_{\left(x,y\right)}\right]\ln\left\{\frac{\mathrm{tr}\;\left[\rho {\left(A\circ B\right)}_{\left(x,y\right)}\right]}{\mathrm{tr}\;\left[{\left(A\circ B\right)}_{\left(x,y\right)}\right]}\right\}\\ \; & =-\sum_{x,y}\mathrm{tr}\;\left[\rho I_{x}^{*}\left(B_{y}\right)\right]\ln\left\{\frac{\mathrm{tr}\;\left[\rho I_{x}^{*}\left(B_{y}\right)\right]}{\mathrm{tr}\;\left[I_{x}^{*}\left(B_{y}\right)\right]}\right\}\\ \; & =-\sum_{x,y}\mathrm{tr}\;\left[I_{x}\left(\rho \right)B_{y}\right]\ln\left\{\frac{\left[I_{x}\left(\rho \right)B_{y}\right]}{\mathrm{tr}\;\left[I_{x}\left(I\right)B_{y}\right]}\right\} \end{array}$$
If $L^{A}$ is the Lüders instrument $I_{x}^{A}\left(\rho \right)=A_{x}^{1/2}\rho A_{x}^{1/2}$ we have ${\left(A\circ B\right)}_{\left(x,y\right)}=A_{x}^{1/2}B_{y}A_{x}^{1/2}$ and
$$S_{A\circ B}\left(\rho \right)=-\sum_{x,y}\mathrm{tr}\;\left(A_{x}^{1/2}\rho A_{x}^{1/2}B_{y}\right)\ln\left[\frac{\mathrm{tr}\;(A_{x}^{1/2}\rho A_{x}^{1/2}B_{y})}{\mathrm{tr}\;(A_{x}B_{y})}\right]$$
If $H^{\left(A,\alpha \right)}$ is the Holevo instrument $H_{x}^{\left(A,\alpha \right)}\left(\rho \right)=\mathrm{tr}\;\left(\rho A_{x}\right)\alpha_{x}$, $\alpha_{x}\in S\left(H\right)$ we obtain
$$\begin{array}{cc} S_{A\circ B}\left(\rho \right)\; & =-\sum_{x,y}\mathrm{tr}\;\left(\rho A_{x}\right)\mathrm{tr}\;\left(\alpha_{x}B_{y}\right)\ln\left[\frac{\mathrm{tr}\;\left(\rho A_{x}\right)\mathrm{tr}\;\left(\alpha_{x}B_{y}\right)}{\mathrm{tr}\;\left(A_{x}\right)\mathrm{tr}\;\left(\alpha_{x}B_{y}\right)}\right]\\ \; & =-\sum_{x,y}\mathrm{tr}\;\left(\rho A_{x}\right)\mathrm{tr}\;\left(\alpha_{x}B_{y}\right)\ln\left[\frac{\mathrm{tr}\;(\rho A_{x})}{\mathrm{tr}\;(A_{x})}\right]\\ \; & =-\sum_{x}\mathrm{tr}\;\left(\rho A_{x}\right)\ln\left[\frac{\mathrm{tr}\;(\rho A_{x})}{\mathrm{tr}\;(A_{x})}\right]=S_{A}\left(\rho \right) \end{array}$$
This also follows from Corollary 8 because
$$\frac{\mathrm{tr}\;(\rho A_{x}\circ B_{y})}{\mathrm{tr}\;(A_{x}\circ B_{y})}=\frac{\mathrm{tr}\;\left(\alpha_{x}B_{y}\right)\mathrm{tr}\;\left(\rho A_{x}\right)}{\mathrm{tr}\;\left(\alpha_{x}B_{y}\right)\mathrm{tr}\;\left(A_{x}\right)}=\frac{\left(\rho A_{x}\right)}{\mathrm{tr}\;(A_{x})}$$

If *A* is an observable on *H* and *B* is an observable on *K* we form the *tensor product observable*A⊗B on H⊗K given by ${\left(A⊗B\right)}_{\left(x,y\right)}=A_{x}⊗B_{y}$ where $\Omega_{A⊗B}=\Omega_{A}\times \Omega_{B}$ [12].

**Lemma 1.** 
*If $\rho_{1}\in S\left(H\right)$, $\rho_{2}\in S\left(K\right)$, then*

$$S_{A\circ B}\left(\rho_{1}⊗\rho_{2}\right)=S_{A}\left(\rho_{1}\right)+S_{B}\left(\rho_{2}\right)$$



**Proof.** From the definition of A⊗B we obtain
$$\begin{array}{cc} S_{A⊗B}\left(\rho_{1}⊗\rho_{2}\right) & =-\sum_{x,y}\mathrm{tr}\;\left(\rho_{1}⊗\rho_{2}A_{x}⊗B_{y}\right)\ln\left[\frac{\mathrm{tr}\;(\rho_{1}⊗\rho_{2}A_{x}⊗B_{y})}{\mathrm{tr}\;(A_{x}⊗B_{y})}\right]\\ \; & =-\sum_{x,y}\mathrm{tr}\;\left(\rho_{1}A_{x}\right)\mathrm{tr}\;\left(\rho_{2}B_{y}\right)\ln\left[\frac{\mathrm{tr}\;\left(\rho_{1}A_{x}\right)\mathrm{tr}\;\left(\rho_{2}B_{y}\right)}{\mathrm{tr}\;\left(A_{x}\right)\mathrm{tr}\;\left(B_{y}\right)}\right]\\ \; & =-\sum_{x,y}\mathrm{tr}\;\left(\rho_{1}A_{x}\right)\mathrm{tr}\;\left(\rho_{2}B_{y}\right)\ln\left[\frac{\mathrm{tr}\;(\rho_{1}A_{x})}{\mathrm{tr}\;(A_{x})}\right]\\ \; & \;-\sum_{x,y}\mathrm{tr}\;\left(\rho_{1}A_{x}\right)\mathrm{tr}\;\left(\rho_{2}B_{y}\right)\ln\left[\frac{\mathrm{tr}\;(\rho_{2}B_{y})}{\mathrm{tr}\;(B_{y})}\right]\\ \; & =-\sum_{x}\mathrm{tr}\;\left(\rho_{1}A_{x}\right)\ln\left[\frac{\mathrm{tr}\;(\rho_{1}A_{x})}{\mathrm{tr}\;(A_{x})}\right]-\sum_{y}\mathrm{tr}\;\left(\rho_{2}B_{y}\right)\ln\left[\frac{\mathrm{tr}\;(\rho_{2}B_{y})}{\mathrm{tr}\;(B_{y})}\right]\\ \; & =S_{A}\left(\rho_{1}\right)+S_{B}\left(\rho_{2}\right)e \end{array}$$ □

We conclude that *A* gives more information about $\rho_{1}$ than *A* and *B* give about $\rho_{1}⊗\rho_{2}$ and similarly for *B*.

A *measurement model* [2,3,9] is a 5-tuple M=(H,K,ν,σ,P) where *H* is the *system* Hilbert space, *K* is the *probe* Hilbert space, ν is the *interaction* channel, σ∈S(K) is the initial *probe state* and *P* is the *probe observable* on *K*. We interpret M as an apparatus that is employed to measure an instrument and hence an observable. In fact, M measures the unique instrument I on *H* given by
$$I_{x}\left(\rho \right)=\mathrm{tr}{\;}_{K}\left[\nu \left(\rho ⊗\sigma \right)\left(I⊗P_{x}\right)\right]$$
In this way, a state ρ∈S(H) is input into the apparatus and combined with the initial state σ of the probe system. The channel ν interacts the two states and a measurement of the probe *P* is performed resulting in outcome *x*. The outcome state is reduced to *H* by applying the partial trace over *K*. Now I measures an unique observable *A* on *H* that satisfies
(4)$$\mathrm{tr}\;\left(\rho A_{x}\right)=\mathrm{tr}\;\left[I_{x}\left(\rho \right)\right]=\mathrm{tr}\;\left[\nu \left(\rho ⊗\sigma \right)\left(I⊗P_{x}\right)\right]$$
The ρ-entropy of I becomes
$$S_{I}\left(\rho \right)=S_{A}\left(\rho \right)=-\sum_{x}\mathrm{tr}\;\left(\rho A_{x}\right)\ln\left[\frac{\mathrm{tr}\;(\rho A_{x})}{\mathrm{tr}\;(A_{x})}\right]$$
where $\mathrm{tr}\;(\rho A_{x})$ is given by (Equation 4). Of course, $S_{I}\left(\rho \right)=S_{A}\left(\rho \right)$ gives the amount of information that a measurement by M provides about ρ. A closely related concept is the observable I⊗P and $S_{I⊗P}\left[\nu (\rho ⊗\sigma )\right]$ also provides the amount of information that a measurement M provides about ρ. It follows from (Equation 4) that the distribution of *A* in the state ρ equals the distribution of I⊗P in the state ν(ρ⊗σ). We now compare $S_{A}\left(\rho \right)$ and $S_{I⊗P}\left[\nu (\rho ⊗\sigma )\right]$. Applying (Equation 4) gives
$$\begin{array}{cc} S_{I⊗P} & \left[\nu (\rho ⊗\sigma )\right]\\ \; & =-\sum_{x}\mathrm{tr}\;\left[\nu \left(\rho ⊗\sigma \right)\left(I⊗P_{x}\right)\right]\ln\left\{\frac{\mathrm{tr}\;\left[\nu \left(\rho ⊗\sigma \right)\left(I⊗P_{x}\right)\right]}{\mathrm{tr}\;(I⊗P_{x})}\right\}\\ \; & =-\sum_{x}\mathrm{tr}\;\left(\rho A_{x}\right)\ln\left[\frac{\mathrm{tr}\;(\rho A_{x})}{n\mathrm{tr}\;(P_{x})}\right]=-\sum_{x}\mathrm{tr}\;\left(\rho A_{x}\right)\ln\left[\frac{\mathrm{tr}\;(A_{x})}{n\mathrm{tr}\;(P_{x})}\;\frac{\mathrm{tr}\;(\rho A_{x})}{\mathrm{tr}\;(A_{x})}\right]\\ \; & =-\sum_{x}\mathrm{tr}\;\left(\rho A_{x}\right)\ln\left[\frac{\mathrm{tr}\;(\rho A_{x})}{\mathrm{tr}\;(A_{x})}\right]-\sum \mathrm{tr}\;\left(\rho A_{x}\right)\ln\left[\frac{\mathrm{tr}\;(A_{x})}{n\mathrm{tr}\;(P_{x})}\right]\\ \; & =S_{A}\left(\rho \right)-\sum_{x}\mathrm{tr}\;\left(\rho A_{x}\right)\ln\left[\frac{\mathrm{tr}\;(A_{x})}{n\mathrm{tr}\;(P_{x})}\right] \end{array}$$
It follows that $S_{A}\left(\rho \right)\leq S_{I⊗P}\left[\nu (\rho ⊗\sigma )\right]$ if and only if
(5)$$\sum_{x}\mathrm{tr}\;\left(\rho A_{x}\right)\ln\left[\frac{\mathrm{tr}\;(A_{x})}{n\mathrm{tr}\;(P_{x})}\right]\leq 0$$
Now (Equation 5) may or may not hold depending on *A*, ρ and *P*. In many cases, *P* is atomic [2,9] and then
$$\ln\left[\frac{\mathrm{tr}\;(A_{x})}{n\mathrm{tr}\;(P_{x})}\right]=\ln\left[\frac{\mathrm{tr}\;(A_{x})}{n}\right]<0$$
so $S_{A}\left(\rho \right)\leq S_{I⊗P}\left[\nu (\rho ⊗\sigma )\right]$ for all ρ∈S(H). Also, (Equation 5) holds if *P* is sharp.

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
