# Peer review of "Entropy of Quantum Measurements"

_entropy, 2022, doi:10.3390/e24111686_

Round 1
Reviewer 1 Report
The author briefly reviews the properties of von Neumann entropy $S$ of the density matrix $\rho$ of a quantum system. Then he uses the notion of quantum effect $a$ in order to define $S_a(\rho)$, which is the $\rho$-entropy of $a$, in Eq.(2.1). The latter quantity $S_a(\rho)$ is interpreted as the amount of uncertainty that the system is in state $\rho$ resulting from a measurement of $a$. In particular, the smaller $S_a(\rho)$ is, the more information a measurement of $a$ gives about $\rho$. Next, the author provides important bounds for $\rho$-entropies of coherent sum of effects or convex mixtures of effects. Finally, he investigates the $\rho$-entropy of any quantum observable $A$. The presented theorems could be useful for practical tasks such as state discrimination.
Overall, I think that this is an excellent contribution and highly recommend its publication.
Author Response
Thank you for your review.
best,
Stan Gudder
Reviewer 2 Report
The topic of the manuscript is relevant to the journal. The manuscript is written rigorously. The structure is well-thought. In Sec. 2, the author introduces a series of theorems on the entropy of effects, which provides the amount of uncertainty that the measurement of the effect gives about the state. Then, in Sec. 3, the notion is employed to define the entropy of observables and instruments. The results are of mathematical nature and the paper lacks a deeper discussion about their physical significance. In particular, the author should address the following points before the manuscript can be published.
1. Please discuss the significance of the results in relation to quantum state tomography, where POVMs are used to reconstruct the density matrix.
2. Can we employ the entropy of observables to compare two incomplete sets of measurements in terms of their efficiency in state tomography?
3. The paper lack clarity when it comes to the definition of the entropy of observables. It is particularly evident in p. 11, where there are several formulas for S_A (\rho) presented in an unorganized manner. The author should first define this concept in the most general way (perhaps, by using \begin{Definition} ... \end{Definition} for better clarity). Then, the special cases, where the formula is simplified, should be enumerated with a clear distinction from the general definition.
4. The manuscript does not contain any concluding section. By adding this part the author can summarize their findings and comment on their significance.
In summary, I consider the manuscript valuable for mathematical physics. However, at this moment, I recommend major revision. If the above comments are properly addressed, the manuscript can be reconsidered for publication.
Author Response

(The authors gave the same response as above.)

Reviewer 3 Report
I very highly recommend this paper for publication in Entropy "as is". It contains fine and interesting findings that are presented and developed nicely.
The paper is methodologically correct.
Publish without delay.
One potential suggestion: maybe the Author should define or explain shortly what he means by a "quantum instruments" (Ref. 14).
Author Response

(The authors gave the same response as above.)
